**Data Availability Statement:** Data cannot be shared publicly because they are PHI. Data are available from the Geisinger Institutional Data

# Associations of four indexes of social determinants of health and two community typologies with new onset type 2 diabetes across a diverse geography in Pennsylvania

Brian S. Schwartz[1,2,3,4]*, Marynia Kolak[5], Jonathan S. Pollak[1], Melissa N. Poulsen[4], Karen Bandeen-Roche[6], Katherine A. Moon[1], Joseph DeWalle[4], Karen R. Siegel[7], Carla I. Mercado[7], Giuseppina Imperatore[7], Annemarie G. Hirsch[1,4]

1 Department of Environmental Health and Engineering, Johns Hopkins Bloomberg School of Public Health, Baltimore, Maryland, United States of America, 2 Department of Epidemiology, Johns Hopkins Bloomberg School of Public Health, Baltimore, Maryland, United States of America, 3 Department of Medicine, Johns Hopkins School of Medicine, Baltimore, Maryland, United States of America, 4 Department of Population Health Sciences, Geisinger, Danville, Pennsylvania, United States of America, 5 Center for Spatial Data Science, University of Chicago, Chicago, Illinois, United States of America, 6 Department of Biostatistics, Johns Hopkins Bloomberg School of Public Health, Baltimore, Maryland, United States of America, 7 Division of Diabetes Translation, National Center for Chronic Disease Prevention and Health Promotion, Centers for Disease Control and Prevention, Atlanta, Georgia, United States of America

* bschwar1@jh.edu

## Abstract

Evaluation of geographic disparities in type 2 diabetes (T2D) onset requires multidimensional approaches at a relevant spatial scale to characterize community types and features that could influence this health outcome. Using Geisinger electronic health records (2008–2016), we conducted a nested case-control study of new onset T2D in a 37-county area of Pennsylvania. The study included 15,888 incident T2D cases and 79,435 controls without diabetes, frequency-matched 1:5 on age, sex, and year of diagnosis or encounter. We characterized patients' residential census tracts by four dimensions of social determinants of health (SDOH) and into a 7-category SDOH census tract typology previously generated for the entire United States by dimension reduction techniques. Finally, because the SDOH census tract typology classified 83% of the study region's census tracts into two heterogeneous categories, termed rural affordable-like and suburban affluent-like, to further delineate geographies relevant to T2D, we subdivided these two typology categories by administrative community types (U.S. Census Bureau minor civil divisions of township, borough, city). We used generalized estimating equations to examine associations of 1) four SDOH indexes, 2) SDOH census tract typology, and 3) modified typology, with odds of new onset T2D, controlling for individual-level confounding variables. Two SDOH dimensions, higher socioeconomic advantage and higher mobility (tracts with fewer seniors and disabled adults) were independently associated with lower odds of T2D. Compared to rural affordable-like as the reference group, residence in tracts categorized as extreme poverty (odds ratio [95% confidence interval] = 1.11 [1.02, 1.21]) or multilingual working (1.07 [1.03, 1.23]) were associated with higher odds of new onset T2D. Suburban affluent-like was associated

Access / Ethics Committee (contact Shannon
Wood, Director, Office of Sponsored Projects, 100
N. Academy Avenue, MC 30-69, Danville, PA,
17822, sewood@geisinger.edu, 570-214-7021) for
researchers who meet the criteria for access to
confidential data.

**Funding:** This work was made possible by
Cooperative Agreement Number U01 DP006296
funded by the U.S. Centers for Disease Control and
Prevention, Division of Diabetes Translation (Co-
PIs AGH and BSS). Three co-authors (KRS, CIM,
GI) are CDC employees and had input into
preparation of the manuscript.

**Competing interests:** The authors have declared
that no competing interests exist.

with lower odds of T2D (0.92 [0.87, 0.97]). With the modified typology, the strongest association (1.37 [1.15, 1.63]) was observed in cities in the suburban affluent-like category (vs. rural affordable-like–township), followed by cities in the rural affordable-like category (1.20 [1.05, 1.36]). We conclude that in evaluating geographic disparities in T2D onset, it is beneficial to conduct simultaneous evaluation of SDOH in multiple dimensions. Associations with the modified typology showed the importance of incorporating governmentally, behaviorally, and experientially relevant community definitions when evaluating geographic health disparities.

## Introduction

Separate from the characteristics of individuals within communities, many features of communities can support or hinder health behaviors and outcomes [1–4]. These features include natural, built, social, economic, cultural, policy, and political characteristics. Among the most important are those that have been broadly termed social determinants of health (SDOH) [5–7], many of which have been studied in relation to geographic or environmental disparities [2, 8]. A large public health literature on community characteristics and health has reported on childhood obesity [9–11], and many of these are also relevant to type 2 diabetes (T2D) [12]. Using different approaches, investigators have operationalized and measured multiple domains of community features aimed at capturing access to or availability of food, physical activity opportunities, utilitarian physical activity (i.e., walkability), socioeconomic advantage or deprivation, and the natural environment (parks, greenness), all of which could influence risk of obesity and T2D in communities through a variety of pathways, including stress, mental health, and diet and physical activity [13–17]. T2D is strongly linked to obesity, is one of the leading causes of adult morbidity and mortality in the U.S., and is a growing public health concern among youth [18].

Specific community factors do not influence health in isolation, but rather interact to create complex risk environments. This multi-dimensional complexity of community factors—coupled with a large range of variability across communities, differential co-occurrence of features within communities, and spatial dependence—presents challenges for studying multiple community features simultaneously. Increasingly, investigators are using a variety of latent variable or cluster analysis models to identify subgroups of communities that incorporate multiple features into **community typologies** [4, 19, 20]. Such techniques allow unique combinations of features and their varying levels within subgroups of communities to be characterized.

*Healthy People 2020* broadly divided SDOH into five key domains: economic opportunity and stability, neighborhoods and built environments, education, social and community context, and health and health care [21]. Each of these five domains can be measured with several different indicators; for example, the economic domain can be evaluated with employment, poverty, food insecurity, and housing instability [22–24]. Many prior public health studies of the economic domain have utilized deprivation indexes based on these indicators [25–28] while ignoring other domains. Recently, Kolak et al. [29] addressed the multidimensional aspects of SDOH by using 15 indicators of SDOH in census tracts across the continental U.S. to derive four indexes (socioeconomic advantage, limited mobility, urban core opportunity, mixed immigrant cohesion and accessibility) using principal components analysis (PCA). Analysis using k-means clustering of the principal components from the PCA [30] yielded a 7-level SDOH census tract typology. This SDOH census tract typology was associated with

measures of premature mortality in Chicago, Illinois. The authors concluded that the typology better captured the complexity and spatial heterogeneity underlying SDOH than when indicators or indexes were used one at a time.

T2D is characterized by strong geographic disparities and spatial heterogeneity [31–34]. We previously reported that in central and northeast Pennsylvania, T2D risk existed at multiple scales, ranging from counties to administrative community types. The latter are derived from U.S. Census Bureau minor civil divisions as township, borough, and city [17, 35]. Minor civil divisions are primary governmental or administrative subdivisions of counties used by 28 states that are governmentally, behaviorally, and experientially relevant, offering a holistic view of community as an organized whole that appears to capture multidimensionality well. We also found that higher community socioeconomic deprivation and lower community greenness [36] were each associated with increased odds of new onset T2D [37]. Herein we extend this work by evaluating Kolak and colleagues' four SDOH principal components and SDOH census tract typology in relation to new onset T2D. While numerous prior studies have observed associations between neighborhood socioeconomic status and diabetes incidence [12], this approach of integrating multiple dimensions of SDOH has not been previously applied in relation to diabetes onset [4].

The study had several specific goals. First, we evaluated how a national census tract typology, which is more amenable to cross-study comparisons than more regional typologies, would classify census tracts and residential locations in a 37-county study region in central and northeastern Pennsylvania. Second, we compared community typology categorizations to administrative community types. Third, given our previous findings that community socioeconomic deprivation in administrative community types was associated with T2D onset [37], we evaluated whether other SDOH were associated with T2D onset. Fourth, we evaluated the SDOH census tract typology in relation to T2D onset. Finally, as the SDOH census tract typology was derived nationally by census tract, a large and less relevant geographic scale especially in rural areas, and we found its use in our region to be less discerning, we combined the SDOH census tract typology with minor civil divisions [17, 35, 38] to compare these different geographies alone and in combination.

## Methods

### Context of study

This study was conducted by Geisinger-Johns Hopkins Bloomberg School of Public Health, one of four academic research centers in the Diabetes LEAD (Location, Environmental Attributes, and Disparities) Network (http://diabetesleadnetwork.org/), dedicated to providing scientific evidence to develop targeted interventions and policies to prevent T2D and related health outcomes across the U.S. The study was approved by the Geisinger Institutional Review Board through a waiver of consent.

### Study population and design

The study population and design have been previously reported [37]. In brief, study participants resided in 37 counties in central and northeastern Pennsylvania and received health care from Geisinger (Fig 1). In a case-control analysis nested within the open dynamic cohort that Geisinger patients represent, Geisinger electronic health records (EHR) were used to identify new onset T2D cases and persons without T2D as controls, from 2008 to 2016, and to define important confounding variables (age, sex, race, ethnicity, Medical Assistance status for health insurance as a surrogate for family socioeconomic status [39], and tobacco use).

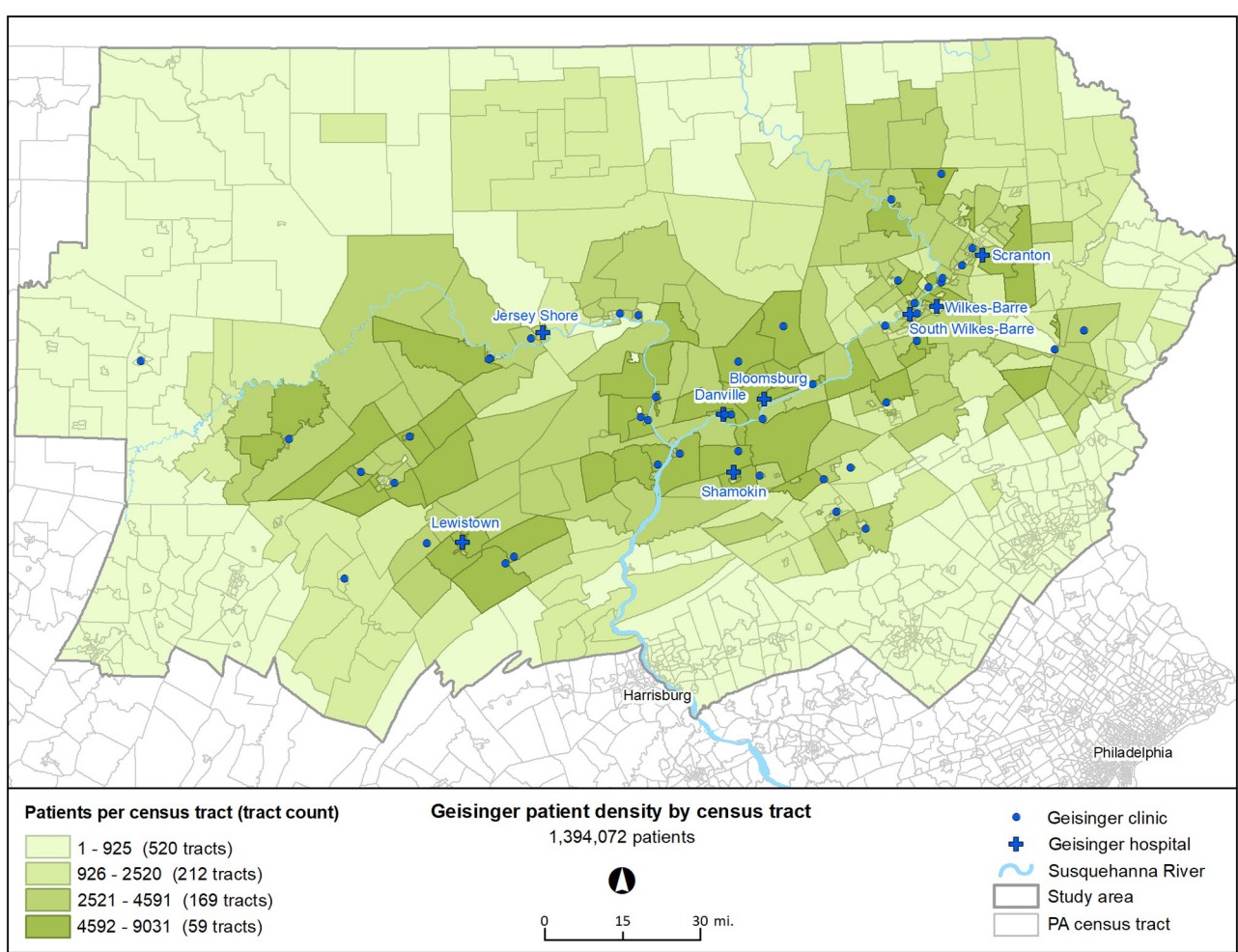

**Fig 1. Distribution of Geisinger patients in the study region.** The study used a July 2017 electronic health record data pull for all ages (n = 1,394,072). Data are displayed for all ages, by census tract, with the number of patients per census tract and the number of tracts in each category. Individuals in the case-control study (n = 95,323) were selected from among these patients > 10 years of age.

Study individuals' addresses at last contact with the health system were obtained through the EHR and geocoded using ArcGIS version 10.4 (ESRI Inc., Redlands, CA). We used census boundaries for minor civil divisions (townships, boroughs, and cities) to define "administrative community types" [35, 37]. We used this approach as rural census tracts can be very large while city minor civil divisions are frequently heterogeneous regarding underlying community characteristics. Our approach subdivided both of these further, so as to reduce within community variability while maximizing between community differences. In the study region, townships range from agriculturally focused rural areas to low density suburbs; boroughs are generally walkable small towns of 5,000 to 10,000 persons with a core area of gridded street networks; and cities are medium-sized urban areas. Residential addresses were then assigned to administrative community types and census tracts. We also classified residential addresses using Census Bureau urbanized area, urban cluster, and rural designations [37]. The study participants resided in 633 townships, 291 boroughs, and 22 cities, representing a total of 785 census tracts.

## Identification of persons with new onset T2D and selection of controls

Persons with new onset T2D (n = 15,888) were identified using an EHR algorithm that relied on encounter diagnoses, diabetes medication orders, and diabetes-relevant laboratory test results [37]. Controls were randomly selected with replacement across years (only one selection per year) and frequency-matched to cases (5:1) on age, sex, and year of encounter date for cases or selection date for controls. Controls (n = 79,435, with 65,084 unique persons) were persons who did not meet any diabetes criteria. To ensure diabetes diagnoses were new onset, we required cases and controls to have at least one encounter with the health system at least two years prior to diagnosis or control selection, without any evidence of diabetes. We also required at least two encounters on different days with a primary care provider (internal medicine, family medicine, pediatrics, or obstetrics and gynecology) prior to the T2D diagnosis or control selection date to ensure that the algorithm would capture T2D diagnosis if present.

## Social determinants of health indexes, census tract typology, and modified typology

We used indexes derived from principal component analysis and an SDOH census tract typology from a previous study that estimated multiple dimensions of SDOH for U.S. census tracts [29]. In this study, fifteen variables from American Community Suvey 2015 data (5-year estimates) were chosen to capture small-area variations in economic status, social and neighborhood characteristics, housing and transportation availability, and demographic characteristics of vulnerable groups and that were also common to multiple SDOH frameworks representing social, economic, and physical environments [29]. These were derived for 71,901 continental U.S. census tracts and then subjected to a PCA that identified four orthogonal principal components that accounted for 71% of the variance in the 15 indicators. The **SDOH principal components** were labeled as 1) the socioeconomic advantage index (accounting for 40.0% of total variance), 2) the limited mobility index (13.4%), 3) the urban core opportunity index (9.6%), and 4) the mixed immigrant cohesion and accessibility index (8.1%) [29]. Finally, an unsupervised, dimension-reducing clustering analysis decomposed census tract into a seven-category **SDOH census tract typology** so that tracts within each category had similar SDOH characteristics based on the four SDOH principal components.

The seven SDOH census tract typology categories identified were complex, and broadly generalized as: rural affordable-like, suburban affluent-like, suburban affordable-like, extreme poverty, multilingual working, urban-like core opportunity, and sparse areas (Table 1) [29]. The terms "rural" and "urban" in these typology labels are not derived from U.S. Census Bureau definitions, but rather termed as approximations capturing broad, nation-wide trends. For example, in our prior work, "rural affordable" tracts can be found on the South Side of Chicago, sharing similar household characteristics of aging, lower income populations found in remote and sparsely populated locales. As such, we add the suffix "-like" to these typology labels in this analysis. Detailed description and additional summary tables can be found in the original manuscript [29]. "Rural affordable-like" areas had higher proportions of older adults and persons with disabilities, moderate income levels, and few persons living in crowded housing or without vehicles, whereas "suburban affluent-like" areas were characterized by high income, a high proportion of children, few households without vehicles, and low poverty. To impose as minimal a construct as possible, a minimum number of qualifiers were used to describe each category. There were high mean (range 535 to 1741) and median (26 to 1187) numbers of persons residing in all census tract typology categories among the 1.39 million persons available in the EHR data that were used for the study (Table 2).

**Table 1. Classification of census tracts and individuals' residential addresses in study area by SDOH census tract typology that was created nationally.**

| Census tract typology class* | Description of census tract typology* | Census tracts | | Cases | | Controls | |
|---|---|---|---|---|---|---|---|
| | | Number | Percent | Number | Percent | Number | Percent |
| Rural affordable-like | High proportions of older adults and persons with disabilities, moderate income levels, and few persons living in crowded housing or without vehicles | 402 | 51.2 | 9701 | 61.1 | 46,274 | 58.3 |
| Suburban affluent-like | High income, high proportion of children, few persons without vehicles, and low poverty | 249 | 31.7 | 4040 | 25.4 | 23,210 | 29.2 |
| Suburban affordable-like | High vehicle ownership, few renters, and high proportion of children | 29 | 3.7 | 547 | 3.4 | 3169 | 4.0 |
| Extreme poverty | High proportion of residents with minority status, low income, high poverty, and high unemployment | 71 | 9.0 | 1061 | 6.7 | 4013 | 5.1 |
| Multilingual working | High proportion of residents with minority status, low English proficiency, low income, and low unemployment | 1 | 0.1 | 13 | 0.1 | 46 | 0.1 |
| Urban-like core opportunity | High income, high proportion of renters, and few children | 16 | 2.0 | 248 | 1.6 | 1469 | 1.9 |
| Sparse areas | High proportion of older adults and generally located within national or state forests, parks, or other natural areas | 17 | 2.2 | 278 | 1.8 | 1254 | 1.6 |
| Total | | 785 | 100.0 | 15,888 | 100.0 | 79,435 | 100.0 |

* These census tract typology names are adopted from [29]. The urban and rural labels do not incorporate other approaches, such as boundaries for urbanized areas or urban clusters.

The SDOH census tract typology classified almost 83% of the 37-county study region's census tracts and over 87% of individuals' residential addresses into two categories: rural affordable-like and suburban affluent-like (Table 1 and Fig 2). A comparison of the SDOH census tract typology with urbanized area, urban cluster, and rural residential address locations showed that 95.2%, 80.57%, and 81.94% of residential addresses in rural, urbanized area, and urban cluster locations, respectively, were in the rural affordable-like or suburban affluent-like community typologies. This motivated the creation of a **modified community typology**, further subdividing the two largest typology categories by **administrative community type**. Of persons residing in rural affordable-like census tracts, 64.7% resided in a township, 29.1% in a borough, and 6.2% in a city. Similarly, of persons residing in suburban affluent-like census tracts, 73.0% resided in a township, 23.3% in a borough, and 3.8% in a city. Of all the residents

**Table 2. Census tracts in region and census tracts and patients in analysis.**

| SDOH Census Tract Typology | Census Tracts, n | | Patients, number per 959 census tracts in 1.39M person EHR data pull* | | Patients, number per 785 census tracts in analysis | |
|---|---|---|---|---|---|---|
| | Total in region** | Total in analysis | Mean (SD) | Median | Mean (SD) | Median |
| Rural affordable-like | 459 (47.9) | 402 (51.2) | 1740.9 (1745.3) | 1184 | 118.5 (162.5) | 51 |
| Suburban affluent-like | 302 (31.5) | 249 (31.7) | 1230.2 (1725.2) | 291 | 91.9 (178.3) | 4 |
| Suburban affordable-like | 38 (4.0) | 29 (3.7) | 1529.3 (2177.5) | 354 | 110.4 (172.7) | 6 |
| Extreme poverty | 115 (12.0) | 71 (9.0) | 934.8 (1373.5) | 152 | 62.0 (91.8) | 25 |
| Multilingual working | 4 (0.4) | 1 (0.1) | 534.5 (1029.0) | 25.5 | 51.0 (NA) | 51 |
| Urban-like core opportunity | 20 (2.1) | 16 (2.0) | 1423.7 (1433.5) | 1187 | 91.1 (103.4) | 53.5 |
| Sparse areas | 21 (2.2) | 17 (2.2) | 1298.2 (1551.9) | 564 | 77.9 (121.5) | 67 |
| Total | 959 (100.0) | 785 (100.0) | 1453.7 (1727.8) | 633 | 103.1 (161.9) | 30 |

* This is the data pull from which the diabetes case-control analysis was designed, obtained in July 2017.

** 37 counties.

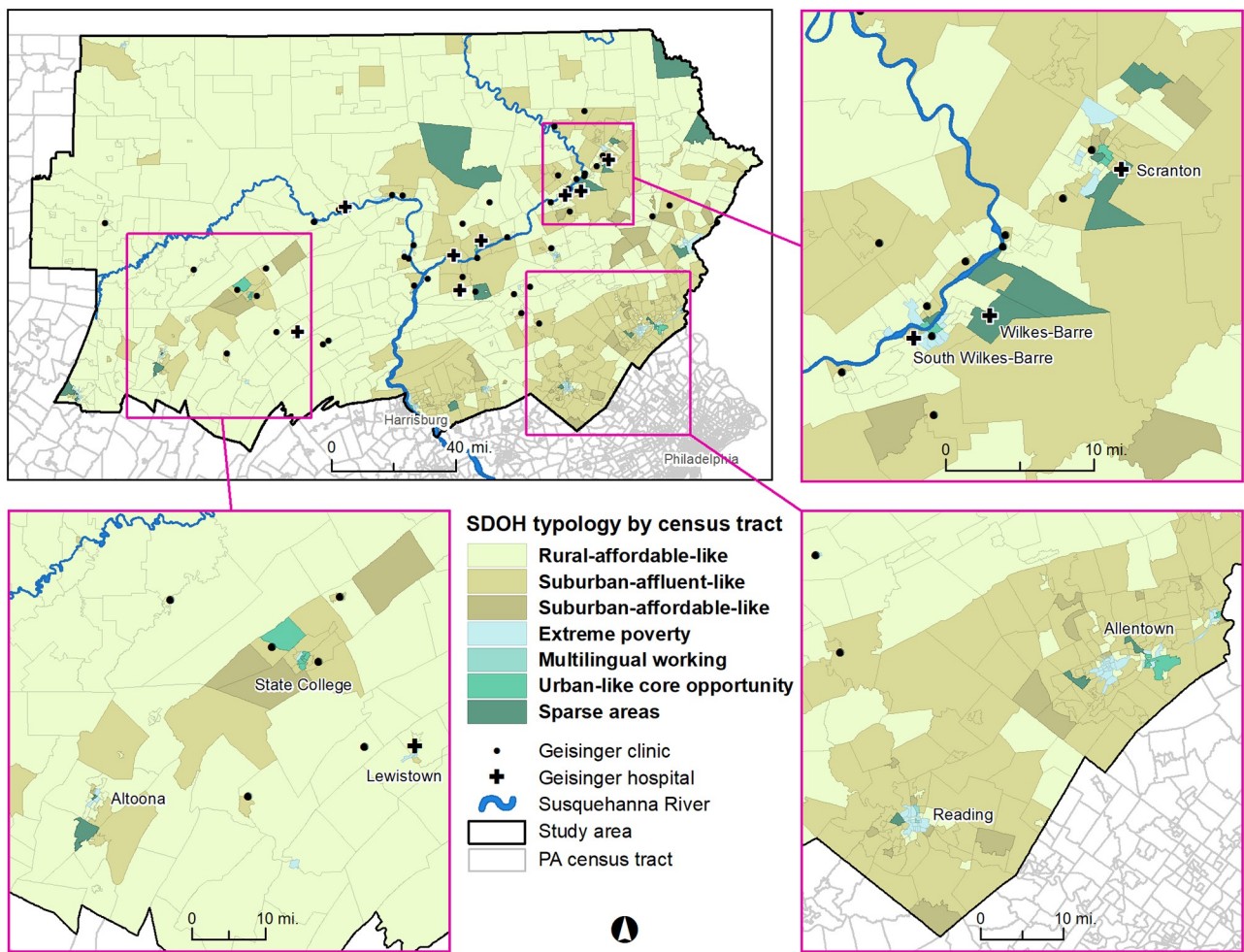

**Fig 2. SDOH census tract typology in 37-county study region.** There were seven typology categories ("SDOH typology by census tract" in legend). The categorization of census tracts used Jenks optimization [40], which divided data into groups to minimize within class variance and maximize between class variance.

who resided in boroughs and cities, 61.8% and 41.3% were categorized as rural affordable-like census tracts (Table 3). T2D and comparison subjects resided in 785 census tracts, 82% of the 959 census tracts in the 37 counties (Table 3). Individuals in the analysis included residents of only one multilingual working census tract among the four in the region.

## Statistical analysis

The goals of the analysis were to: (1) categorize and describe census tracts and residential locations within a specific region (37 counties within a single state) using a nationally-generated census tract typology; (2) compare the census tract typology classification to administrative community types (townships, boroughs, and cities), a very relevant context that is associated with many health outcomes in the study region; (3) evaluate associations of the four SDOH principal components [29], alone and in combination, in relation to T2D onset to evaluate whether these lead to additional insights compared to our prior finding using community socioeconomic deprivation alone [37]; (4) evaluate associations of the SDOH census tract typology with new onset T2D; and (5) evaluate associations of a modified typology (combining

**Table 3. Classification of individuals' residential addresses by administrative community types and SDOH census tract typology.**

| Typology class | Administrative Community Types | | | | | |
|---|---|---|---|---|---|---|
| | Boroughs | | Cities | | Townships | |
| | Number | Percent | Number | Percent | Number | Percent |
| Rural affordable-like | 16,289 | 61.8 | 3446 | 41.3 | 36,240 | 59.8 |
| Suburban affluent-like | 6346 | 24.1 | 1025 | 12.3 | 19,879 | 32.8 |
| Suburban affordable-like | 298 | 1.1 | 209 | 2.5 | 3209 | 5.3 |
| Extreme poverty | 1844 | 7.0 | 3205 | 38.4 | 25 | 0.04 |
| Multilingual working | 0 | 0.0 | 51 | 0.6 | 0 | 0.0 |
| Urban-like core opportunity | 1243 | 4.7 | 150 | 1.8 | 324 | 0.5 |
| Sparse areas | 357 | 1.4 | 260 | 3.1 | 915 | 1.5 |
| Total | 26,377 | 100.0 | 8354 | 100.0 | 60,592 | 100.0 |

administrative community types with the SDOH census tract typology) with new onset T2D. Analysis of the four SDOH principal components allows capture of distinct social and economic trends on a continuous scale, while the community typology organizes tracts into seven categories of these trends.

The approach to the analysis has been previously reported [37]. Univariate distributions for study variables were first examined. Key study variables were then described within community definitions of interest. Statistical comparison of case and control characteristics accounted for repeated control selection. In unadjusted comparisons of cases and controls, probability weighting for the number of appearances in the dataset was applied for analyses of time-invariant characteristics, and mixed-effects models with person-specific intercepts were applied for analyses of time-varying characteristics, depending on the measurement scale. Statistical analysis was completed using Stata-MP version 15.1 (StataCorp LLC, College Station, TX).

Logistic regression was used to estimate associations (odds ratios, 95% confidence intervals) using generalized estimating equations with robust standard errors and an exchangeable correlation structure within administrative community types while adjusting for confounding variables. We adjusted for age in years (linear, with quadratic and cubic terms to allow for nonlinearity), sex, race (White vs. all other races), ethnicity (Hispanic vs. non-Hispanic), and Medical Assistance use while under care at Geisinger (a surrogate for family socioeconomic status, as > 0% of time vs. 0%) [39]. Information on race was obtained from the EHR. Race was included in models as a social construct that can cause, or is correlated with, many factors that can affect health, such as racism, discrimination, and many SDOH. Race was initially evaluated in six categories (White, African American, Asian, Native American, Pacific Islander or Hawaiian, mixed), but dichotomized after associations with community types did not change based on the race parameterization. Other variables in the EHR, such as body mass index (BMI) and tobacco use, were not included in the analysis because these were considered to be mediators (BMI) or not confounders (tobacco).

Each of the SDOH principal components were evaluated separately and then in models together. The 7-category SDOH census tract typology and the modified typology were then evaluated without SDOH principal components in the models. We used administrative community type for the modified typology, which combined minor civil divisions outside of cities (townships and boroughs) with census tracts within cities, because it is a more relevant context than census tract and because of limitations of the nationally derived SDOH census tract typology when applied regionally. In the absence of minor civil division boundaries, which are not available nationally, a similar approach could be used across urbanized area, urban cluster, and

rural designations. We followed best spatial analytic practices in selecting the most relevant unit construct to represent lived experiences within our local region.

## Results

### Characteristics of study population and SDOH in census tracts

Characteristics of the study population have been previously reported [37]. They had an average age of 55 years; were predominantly white and non-Hispanic; the majority had a Geisinger primary care provider; and most cases were diagnosed with T2D in an outpatient setting. The correlations among the four SDOH principal components as measured in the study region's 785 census tracts ranged from -0.34 (socioeconomic advantage [PC1] with mixed immigrant cohesion [PC4]) to 0.45 (socioeconomic advantage [PC1] with limited mobility [PC2]). The Pearson's correlation between community socioeconomic deprivation (higher = more deprivation, from [37]) and socioeconomic advantage (PC1, higher = greater advantage) was -0.70.

### Community typology and administrative community types

We further explored the SDOH community typology categorization by subdividing the two largest census tract typology categories (rural affordable-like, suburban affluent-like) by administrative community types, resulting in a modified typology of 11 categories (Table 4; the five less prevalent census tract typology categories were not subdivided). The highest average proportion of non-white individuals resided in census tracts in the suburban affordable typology (8.8%), followed by multilingual working (6.8%) and extreme poverty (5.4%) (Table 4). The highest average proportion of Hispanic individuals resided in census tracts in the multilingual working typology (13.6%) followed by suburban affordable-like (5.8%) and extreme poverty (5.4%). The modified typology demonstrated important and large differences between administrative community types within the same SDOH census tract typology categories. For example, among rural affordable-like census tracts, on average, there were very large differences in city vs. township administrative community types for population density (5959 vs. 150 persons/mi$^2$), percent developed (78.8% vs. 3.8%), mobility (PC2, -1.59 vs. -0.69), community socioeconomic deprivation (2.68 vs. 0.15), greenness (0.52 vs. 0.74), and socioeconomic advantage (PC1, 0.10 vs. 1.84) (Table 4). Similar large differences were observed among administrative community types in the suburban affluent-like category.

### Associations with T2D onset

In SDOH principal component models, higher socioeconomic advantage (PC1) and higher mobility (PC2) (corresponding to areas with fewer seniors and disabled adults) were each associated with lower odds of T2D (**Models 1 and 2**, Table 5). The urban core opportunity (PC3) and mixed immigrant cohesion and accessibility (PC4) indexes were not associated with new onset T2D (**Models 3 and 4**, global test p-values >0.20). When the four principal components were added together in pairs to the base model, there was little substantive change in associations. For example, when PC1 and PC2 were included together in a single model, associations for each were attenuated, but higher levels of both principal components were still associated with lower odds of T2D when modeled as quartiles (**Model 1 + 2**, Table 5) and or as continuous variables (results not shown).

SDOH census tract typology was next evaluated in relation to new onset T2D. Compared to rural affordable-like as the reference group, residence in census tracts categorized as extreme poverty (odds ratio [OR] = 1.11) or multilingual working (OR = 1.07) was associated with higher odds of new onset T2D while suburban affluent-like was associated with lower odds of

**Table 4. Selected summary statistics by modified typology categories among 95,323 cases and controls.**

| Category | Values Among Individuals | | | Values for Administrative Community Types Assigned to Persons | | | | | |
|---|---|---|---|---|---|---|---|---|---|
| | Non-white race, %, mean | Hispanic ethnicity, %, mean | Medical Assistance ≥ 50%, %, mean | Percent developed, mean | Socioeconomic advantage (PC1), mean (SD) | Limited mobility (PC2), mean (SD) | Community socioeconomic deprivation, mean (SD) | Population density, persons/square mile, mean (SD) | Peak NDVI, mean (SD) |
| Rural affordable-like–borough | 1.1 | 1.3 | 5.2 | 49.2 | 0.99 (0.68) | -1.07 (0.59) | 1.05 (2.11) | 2912 (1980) | 0.61 (0.10) |
| Rural affordable-like–city | 4.5 | 4.0 | 8.2 | 78.8 | 0.10 (0.73) | -1.59 (0.70) | 2.68 (1.54) | 5959 (2518) | 0.52 (0.09) |
| Rural affordable-like–township | 1.3 | 0.9 | 3.2 | 3.8 | 1.44 (0.63) | -0.69 (0.59) | 0.15 (2.34) | 150.4 (150.3) | 0.74 (0.10) |
| Suburban affluent-like–borough | 1.6 | 0.8 | 3.4 | 39.5 | 1.84 (0.61) | -0.37 (0.49) | -0.19 (2.71) | 2496 (2184) | 0.64 (0.11) |
| Suburban affluent-like–city | 3.9 | 3.1 | 6.6 | 57.2 | 1.05 (0.93) | -0.55 (0.43) | 0.60 (1.88) | 3897 (2618) | 0.59 (0.09) |
| Suburban affluent-like–township | 1.8 | 0.9 | 1.7 | 6.3 | 2.27 (0.44) | -0.02 (0.50) | -2.00 (1.90) | 266.9 (234.9) | 0.72 (0.09) |
| Suburban affordable-like | 8.8 | 5.8 | 2.7 | 8.8 | 1.29 (1.44) | 0.79 (0.60) | -1.52 (2.58) | 585.5 (1294) | 0.74 (0.11) |
| Extreme poverty | 5.4 | 5.4 | 11.3 | 69.6 | -1.48 (0.97) | -1.73 (0.80) | 4.51 (1.42) | 6115 (3023) | 0.52 (0.11) |
| Multilingual working | 6.8 | 13.6 | 13.6 | 40.8 | -3.96 (0.00) | -0.05 (0.00) | 4.90 (1.59) | 3787 (0.00) | 0.42 (0.06) |
| Urban-like core opportunity | 5.1 | 1.4 | 3.7 | 46.5 | -0.27 (1.04) | -0.72 (0.77) | 2.40 (2.26) | 5228 (3602) | 0.57 (0.12) |
| Sparse areas | 2.1 | 1.0 | 2.9 | 26.8 | 1.54 (0.90) | -2.46 (0.63) | 1.41 (2.70) | 1415 (2036) | 0.65 (0.13) |

Abbreviations: NDVI = normalized difference vegetation index; PC = principal component; SD = standard deviation.

T2D (OR = 0.92) (**Model 5**, Table 5). The modified typology that subcategorized some census tracts by administrative community types seemed to capture important features of residential location, as the range of the strength of associations increased (**Model 6**, Table 5). The highest odds was found among residents in cities in the suburban affluent-like category (OR = 1.37), followed by cities in the rural affordable-like category (OR = 1.20). The associations for extreme poverty (OR = 1.16) and multilingual working (OR = 1.12) also strengthened, as the reference group was not the entire rural affordable-like category, but only the subset in townships.

## Discussion

T2D has strong geographic disparities. The features of communities that may contribute to higher or lower risks of new onset T2D are multidimensional, cluster differently in different types of communities, have highly varied distributions in different types of communities, and likely interact with one another. For example, walkable neighborhoods may not promote physical activity or stress reduction in areas with high proportions of disabled or older residents,

**Table 5. Adjusted\* associations of SDOH principal components, SDOH census tract typology, and modified community typology <u>from separate models</u> with new onset T2D status.**

| Variable | OR (95% CI) |
| --- | --- |
| **SDOH Principal Components** | |
| **Model 1**: Socioeconomic advantage (PC1) | |
| Quartile 1 | 1.0 |
| Quartile 2 | 0.94 (0.86, 1.01) |
| Quartile 3 | 0.90 (0.84, 0.97) |
| Quartile 4 | 0.79 (0.74, 0.85) |
| **Model 2**: Limited mobility index (PC2) | |
| Quartile 1 | 1.0 |
| Quartile 2 | 0.94 (0.88, 1.00) |
| Quartile 3 | 0.90 (0.85, 0.96) |
| Quartile 4 | 0.84 (0.79, 0.91) |
| **Model 3**: Urban core opportunity (PC3) | |
| Quartile 1 | 1.0 |
| Quartile 2 | 0.97 (0.92, 1.03) |
| Quartile 3 | 1.00 (0.93, 1.07) |
| Quartile 4 | 1.001 (0.92, 1.08) |
| **Model 4**: Mixed immigrant cohesion (PC4) | |
| Quartile 1 | 1.0 |
| Quartile 2 | 1.04 (0.98, 1.10) |
| Quartile 3 | 1.03 (0.96, 1.12) |
| Quartile 4 | 1.08 (1.00, 1.16) |
| **Model 1 + 2**: PC1 + PC2 | |
| PC1 –socioeconomic advantage | |
| Quartile 1 | 1.0 |
| Quartile 2 | 0.94 (0.87, 1.02) |
| Quartile 3 | 0.90 (0.84, 0.98) |
| Quartile 4 | 0.82 (0.76, 0.89) |
| PC2 –limited mobility | |
| Quartile 1 | 1.0 |
| Quartile 2 | 0.98 (0.92, 1.04) |
| Quartile 3 | 0.96 (0.90, 1.02) |
| Quartile 4 | 0.91 (0.85, 0.98) |
| **SDOH Census Tract Typology and Modified Typology** | |
| **Model 5**: SDOH census tract typology categories | |
| Rural affordable-like | 1.0 |
| Suburban affluent-like | 0.92 (0.87, 0.97) |
| Suburban affordable-like | 0.95 (0.84, 1.07) |
| Extreme poverty | 1.11 (1.02, 1.21) |
| Multilingual working | 1.07 (1.03, 1.23) |
| Urban-like core opportunity | 1.04 (0.92, 1.18) |
| Sparse areas | 1.06 (0.95, 1.19) |
| **Model 6**: <u>modified</u>† typology categories | |
| Rural affordable-like–township | 1.0 |
| Rural affordable-like–borough | 1.08 (1.02, 1.15) |
| Rural affordable-like–city | 1.20 (1.05, 1.36) |
| Suburban affluent-like–township | 0.92 (0.86 0.98) |

(*Continued*)

**Table 5.** (Continued)

| Variable | OR (95% CI) |
|---|---|
| Suburban affluent-like–borough | 0.94 (0.86, 1.03) |
| Suburban affluent-like–city | 1.37 (1.15, 1.63) |
| Suburban affordable-like | 0.97 (0.85, 1.10) |
| Extreme poverty | 1.16 (1.06, 1.26) |
| Multilingual working | 1.12 (1.07, 1.17) |
| Urban-like core opportunity | 1.06 (0.92, 1.22) |
| Sparse areas | 1.10 (0.98, 1.23) |

* Logistic regression using generalized estimating equations with robust standard errors; one community or community feature variable was in the model at a time; models adjusted for sex, race (White vs. all others), ethnicity (Hispanic vs. non-Hispanic), age (age, $age^2$, $age^3$), and Medical Assistance status (0% vs. > 0% of time).

† The modified typology subdivided the two largest typology categories by administrative community type.

** Quartile cutoffs in the 785 census tracts in the analysis: PC1: < 0.293, 0.293 to 1.400, 1.401 to 1.959, > 1.959; PC2: < -1.179, -1.179 to -0.618, -0.619 to -0.085, > -0.085; PC3: < -0.626, -0.626 to -0.277, -0.278 to 0.105, > 0.105; and PC4: < -0.467, -0.467 to -0.139, -0.140 to 0.320, > 0.320.

high levels of community socioeconomic deprivation, or high crime rates. One approach to these complexities is to define a community typology that simultaneously accounts for these multidimensional and distributional issues, but such an approach had never been used before in relation to new onset T2D. In this first evaluation of community typology with T2D, we extended work that used 15 SDOH indicators to identify four principal components which were then used to identify a seven-level census tract typology [29]. The results supported several conclusions: community typologies were associated with new onset T2D; risk differences were found at small scales (i.e., in this region the area of boroughs and city census tracts averages approximately one square mile) and were higher in urban areas; several dimensions of SDOH were associated with T2D onset; and when census tract typologies were subdivided using a more relevant context in our region, specifically administrative community type, greater variation in T2D onset by community was identified. This study provides additional evidence that T2D risk is identifiable in small geographies and, within this 37-county study region, is higher in urban areas of a variety of types [37].

Previous studies identified higher crude prevalence rates of diabetes in rural areas as compared to urban areas [41], but after adjusting for risk factors found lower prevalence in rural areas [42]. Population-dense urban areas may generally have more resources, transit options, and walkable environments [43] as compared to rural areas, however these benefits supporting healthy behaviors may not consistently translate to improved T2D outcomes. Greater economic disparities and residential segregation may also drive T2D health disparities [44–47], and such phenomena may serve to worsen outcomes when concentrated in urban areas. The interactions of all these features generate a nuanced landscape that may drive different health outcomes in different areas. In this regional setting, areas with more patients diagnosed with T2D were associated with densely populated city tracts that had more development, less vegetation (i.e. greenness using the normalized difference vegetation index), greater socioeconomic disadvantage, and more patients with need of Medical Assistance. Interpretation of health disparities across urban and rural divides thus remains complex, necessitating an improved understanding of the localized, place-based context.

The national typology measurement approach [29] applied to a smaller region showed limited variation in assignments by census tract (i.e., 83% were assigned to two types, and there

was only one multilingual working census tract in the analysis). This suggests that there are likely smaller scale differences in how SDOH are distributed and that future studies could evaluate community typologies on a regional scale. As expected, while the four principal components were uncorrelated (orthogonal) on a national basis, they were not orthogonal regionally. This suggests that SDOH measures may have differential item functioning by region. However, while a regional approach allows for spatial non-stationarity [31] and captures regional measurement differences that are likely to exist, it is sample dependent and less comparable across studies.

The study provided evidence that other dimensions of SDOH were associated with T2D onset, alone and in combination. After socioeconomic advantage, highly correlated with a previously evaluated metric–community socioeconomic deprivation [37]–higher mobility was most strongly associated with lower odds of T2D onset, and the highest quartile of the mixed immigrant cohesion index was also associated with higher odds of new onset T2D. The associations of socioeconomic advantage and the limited mobility index were little changed when the two were included in models together.

These findings support previous work linking different dimensions of social vulnerability with T2D as a risk factor but extend these concepts as dimensions summarized by neighborhood types. Typologies with worse socioeconomic disadvantage in our study area, extreme poverty and multilingual working communities, also had worse T2D outcomes. Neighborhoods with concentrated poverty have co-occurrence of multiple T2D risk factors like poor access to healthy foods, recreational facilities, and health services, further compounding individual risk factors like reduced income [48]. Recent immigrant communities may experience additional barriers to language, health insurance, and social service safety nets [49, 50]. A previous study of T2D at the census tract-level for a clinical health population also found increased incidence in communities with more residents with limited English proficiency and persons without health insurance [51]. Areas with more concentrated seniors and disabled adult populations may also reflect additional challenges between aging, mobility, and transit that serve as risk factors for T2D onset. Aging populations tend to participate in less physical activity [52], which may further drive onset and/or worsen outcomes or severity of disease. At a geographic level, the factors driving the production of aging communities are varied and may also complicate outcomes. Identifying the dimensions of social vulnerability is essential to generating improved accessibility services, meaningful planning policies, and appropriate place-based interventions to improve environments for health.

The combination of a census tract community typology with administrative community type allowed several insights into how geographic disparities in health outcomes could be evaluated. Census tracts are available nationally but have many limitations as a definition of community or neighborhood [35]. In contrast, administrative community types are a useful measure regionally that we have used as an experientially, behaviorally, and governmentally relevant definition of community in several prior studies of a range of environmental and community conditions and health outcomes [36, 53–55]. However, minor civil division boundaries we used for administrative community type are only available for a subset of U.S. states (as of 2010, for 29 states), limiting nation-wide analyses.

There were several limitations to the study. Community typologies were derived in a data-driven approach starting with 15 SDOH indicators available at the tract-level across the country, identified with a data reduction and clustering analysis approach. While these indicators capture some basic elements of SDOH broadly at a national scale, future work evaluating a broader range of indicators, and further be attenuated to localized, region-specific aspects, would be valuable. Some of the SDOH census tract typology categories had few census tracts (e.g., multilingual working had only one), so conclusions about those categories should be

cautious. While we started with individual level EHR records, we identified associations at group-levels across census tracts, and thus reflect population means rather than individual phenomena. Findings are thus subject to the modified areal unit problem [56] and should be interpreted with caution. We focused on patients with diagnosed T2D and do not assess the counterfactuals, like undiagnosed T2D patient outcomes, or patients with T2D that did not utilize services within the observed clinical setting. We focused on one large clinical population, though more refined estimates of healthcare quality received within the population, or health outcomes of residents outside the clinical population, were not available. Healthcare system quality and built environment may both influence health behaviors, and thus impact health disparities. Additional refined measures capturing more variability within the environment like vegetation indexes, metrics reflecting the walkability of areas (e.g., intersection density), local pollution estimation, and resource distribution specific to disease of interest could improve future work.

Future studies should consider evaluating community features simultaneously in the multiple environmental and community domains that are specifically relevant to the dietary behaviors, physical activity, and stress promotion or reduction relevant to obesity and T2D, such as land use, food, physical activity, and social environments and community greenness and blue space. A wide variety of measures is available to characterize each of these so considerable work will be needed to standardize these, select among them, eliminate the highly correlated ones, and evaluate whether there is utility in application of one of several finite mixture models in defining community typologies and identifying geographic concentrations of health risk [4].

## Acknowledgments

The authors thank Dione G. Mercer for project management. The findings and conclusions in this manuscript are those of the authors and do not necessarily represent the official position of the Centers for Disease Control and Prevention.

## Author Contributions

**Conceptualization:** Brian S. Schwartz, Annemarie G. Hirsch.

**Data curation:** Brian S. Schwartz, Jonathan S. Pollak, Joseph DeWalle.

**Formal analysis:** Jonathan S. Pollak, Karen Bandeen-Roche, Joseph DeWalle.

**Funding acquisition:** Brian S. Schwartz, Annemarie G. Hirsch.

**Investigation:** Brian S. Schwartz, Marynia Kolak, Melissa N. Poulsen, Katherine A. Moon, Karen R. Siegel, Carla I. Mercado, Giuseppina Imperatore, Annemarie G. Hirsch.

**Methodology:** Brian S. Schwartz, Marynia Kolak, Jonathan S. Pollak, Melissa N. Poulsen, Karen Bandeen-Roche, Katherine A. Moon, Karen R. Siegel, Carla I. Mercado, Giuseppina Imperatore, Annemarie G. Hirsch.

**Project administration:** Annemarie G. Hirsch.

**Resources:** Marynia Kolak.

**Supervision:** Brian S. Schwartz, Karen Bandeen-Roche, Annemarie G. Hirsch.

**Writing – original draft:** Brian S. Schwartz.

**Writing – review & editing:** Brian S. Schwartz, Marynia Kolak, Jonathan S. Pollak, Melissa N. Poulsen, Karen Bandeen-Roche, Katherine A. Moon, Joseph DeWalle, Karen R. Siegel, Carla I. Mercado, Giuseppina Imperatore, Annemarie G. Hirsch.

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
