## [Decision Letter · Decision Letter 0]

15 Jun 2022

PONE-D-21-12441

Associations of four indices of social determinants of health and two community typologies with new onset type 2 diabetes across a diverse geography in Pennsylvania

PLOS ONE

Dear Dr. Schwartz,

Thank you for submitting your manuscript to PLOS ONE. After careful consideration, we feel that it has merit but does not fully meet PLOS ONE’s publication criteria as it currently stands. Therefore, we invite you to submit a revised version of the manuscript that addresses the points raised during the review process.

We look forward to receiving your revised manuscript.

Kind regards,

Vanessa Carels

Staff Editor

PLOS ONE

Journal Requirements:

5. We note that Figures 1 and 2 in your submission contain map images which may be copyrighted. All PLOS content is published under the Creative Commons Attribution License (CC BY 4.0), which means that the manuscript, images, and Supporting Information files will be freely available online, and any third party is permitted to access, download, copy, distribute, and use these materials in any way, even commercially, with proper attribution. For these reasons, we cannot publish previously copyrighted maps or satellite images created using proprietary data, such as Google software (Google Maps, Street View, and Earth). For more information, see our copyright guidelines: http://journals.plos.org/plosone/s/licenses-and-copyright.

a. You may seek permission from the original copyright holder of Figures 1 and 2 to publish the content specifically under the CC BY 4.0 license.  

Reviewers' comments:

Reviewer's Responses to Questions

**Comments to the Author**

1. Is the manuscript technically sound, and do the data support the conclusions?

Reviewer #1: Yes

Reviewer #2: Yes

2. Has the statistical analysis been performed appropriately and rigorously? 

Reviewer #1: Yes

Reviewer #2: Yes

3. Have the authors made all data underlying the findings in their manuscript fully available?

Reviewer #1: Yes

Reviewer #2: No

4. Is the manuscript presented in an intelligible fashion and written in standard English?

Reviewer #1: Yes

Reviewer #2: Yes

5. Review Comments to the Author

Reviewer #1: The paper is generally well written but need English improvements. The introduction is relevant. about the study findings is not presented or reported well need to focus on the standard reporting the result. The methods are generally appropriate Moreover, the inclusion criteria need more details.

Reviewer #2: Overall: This is a very well done study of incident diabetes risk in a relatively rural community in Pennsylvania. Very careful attention is given by the authors in defining community types very specifically for this rural area of the country. They expertly show how it is not sufficient to use national typologies to describe health outcomes in more local environments. While some might consider the stepwise and detailed defining of community types laborious and over-complicated, I feel that this is a best-of-class demonstration in the geographic literature, which shows why this sort of exacting attention to how we define different local environments is critically necessary not only to point out key associated risk factors, but also so that the correct conclusions can be made about health outcomes and how geography plays a central role in defining risks for poor outcomes.

Abstract:

In the background of abstract, would it make sense to say “associated community types and features”

In the methods of the abstract, that a count of incident cases of diabetes is so it would be helpful to clarify as “15,888 incident T2D cases”

Please clarify what “generated nationally” mean in lines 35-36

It’s not clear what the purpose of: “subdivided the two largest SDOH census tract typology categories by administrative community types (U.S. Census Bureau minor civil divisions of

township, borough, city)” – I presume this is to further delineate geography at a refined level or subtype tracts by urban/rural. It might be better to say the goal and approach in more general terms leaving the specifics to the body of the study.

In the results of the abstract, consider leading with the distribution of tract typologies as a list, e.g., ___ (__%), ___ (__%), ___ (__%), ___ (__%). As written now, it’s difficult to understand what rural affordable-like and suburban affluent-like are being compared to, especially since these categories are respectively 83% and 87% they appear to be in different typologies.

In the conclusion of the abstract, it appears that a main goal of the study is to develop a more refined typology to study the geography of a health outcome. It might be worthwhile to point out this goal up front.

Introduction

Minor civil divisions is described on lines 114-115 as “a holistic view of community as an organized whole that is experientially and behaviorally relevant and appears to capture multidimensionality well” is a bit confusing. In more specific terms what is a minor civil division and what geographic scale could it be compared to? In other words, is this just used in this area of Pennsylvania or can you describe it less abstractly?

Methods

Regarding, this point on line 141-142, “rural census tracts can be very large while city minor civil divisions are frequently heterogeneous regarding underlying community characteristics” – I would think if rural tracts are large then residents within smaller civil divisions would be more homogenous with each other? I think what the authors are trying to say is that civil divisions are more heterogenous from each other? Either way the wording could be clarified here by pointing out civil divisions are actually smaller than tracts and provide better differentiation between communities in rural areas than tracts would.

Noting that there are 7 typologies, which might actually be hard to list out in the abstract as I wrote earlier. Now reading back at the abstract I realize that it was 83% of tracts versus 87% of individuals addresses. Maybe makes sense to just limit the description just to either tracts or individuals, whichever was the primary unit of analysis. Plus dividing and actually reporting % rural affordable-like versus suburban affluent-like and say those were the highest two categories.

Table 2 could use % in the first two columns.

In lines 199 – 201 is there a reason for reporting these two top categories together rather than separately? Are they being analyzed together?

Some description of township versus borough versus city would be helpful in lines 207-208.

In line 246, % time using Medical Assistance sounds continuous, but then 0% versus >0% in line 247 sounds binary. I would just describe it as used.

Did the dichotomization of race materially affect any results? Unclear why not just leave it as is.

Results

Part of me questions the “multilingual working” category which was one of the highest proportion non-Hispanic White. Wondering given the small numbers and attributes of these areas if they really represent the risk among multilingual working individuals or if they become averaged out.

Though arguably the proportion Hispanic is highest in these areas and their high proportion might very well differentiate the non-Hispanic White residents in these areas versus in other areas.

In addition the diabetes risk is quite different in these specific areas. That is probably the most demonstrative finding that shows why one should be doing this sort of detailed typology.

I guess the question is whether the risk is really among the Hispanic proportion of these areas, which is averaged with non-Hispanic White residents in these areas, versus whether it is also elevated risk among the non-Hispanic White residents in these multilingual working individuals that contributes to the risk in these areas.

Since the data includes individual level data on race/ethnicity, is it possible to answer a question like this?

Discussion

The discussion is well balanced and raises several astute points. The one point I might make that counter balances the points raised in lines 342 – 347 is that diabetes in this relatively more rural areas also cluster in the relatively more urban areas within rural counties. The maps in the Geisinger catchment appear to reflect this as well. So then the problem of rural diabetes is also clustered in the more urban parts of rural counties, so then in some ways even the overall problem of rural diabetes might be considered a relatively more urban phenomenon. I think the authors understand this based on lines 348 – 351.

Data Availability

Regarding data availability, authors cite that data is PHI. Furthermore, I would say that in a sparsely populated area such as this, it might be easier to identify individuals even with a de-identified dataset given the rural study population. Therefore, it is probably reasonable for the authors to request that datasets be made available only through explicit DUA.

6. PLOS authors have the option to publish the peer review history of their article (what does this mean?). If published, this will include your full peer review and any attached files.

Reviewer #1: **Yes: **MAHMOUD AL-MASAEED

Reviewer #2: No

---

## [Decision Letter · Decision Letter 1]

4 Sep 2022

Associations of four indexes of social determinants of health and two community typologies with new onset type 2 diabetes across a diverse geography in Pennsylvania

PONE-D-21-12441R1

Dear Dr. Schwartz,

We’re pleased to inform you that your manuscript has been judged scientifically suitable for publication and will be formally accepted for publication once it meets all outstanding technical requirements.

Kind regards,

Celine Rozenblat

Academic Editor

PLOS ONE

Additional Editor Comments (optional):

Reviewers' comments:

Reviewer's Responses to Questions

**Comments to the Author**

1. If the authors have adequately addressed your comments raised in a previous round of review and you feel that this manuscript is now acceptable for publication, you may indicate that here to bypass the “Comments to the Author” section, enter your conflict of interest statement in the “Confidential to Editor” section, and submit your "Accept" recommendation.

Reviewer #1: All comments have been addressed

Reviewer #2: All comments have been addressed

2. Is the manuscript technically sound, and do the data support the conclusions?

Reviewer #1: Yes

Reviewer #2: Yes

3. Has the statistical analysis been performed appropriately and rigorously? 

Reviewer #1: Yes

Reviewer #2: Yes

4. Have the authors made all data underlying the findings in their manuscript fully available?

Reviewer #1: Yes

Reviewer #2: Yes

5. Is the manuscript presented in an intelligible fashion and written in standard English?

Reviewer #1: Yes

Reviewer #2: Yes

6. Review Comments to the Author

Reviewer #1: This is an interesting study and the authors have collected a unique dataset using cutting edge methodology. The paper is generally well written and structured. Overall, this is a clear, concise, and well-written manuscript. The introduction is relevant, Moreover, it would be better if you write more about the findings.

Reviewer #2: Excellent and thoughtful responses to all comments made by the reviewers. Also, agree with their assessment of data access as noted.

7. PLOS authors have the option to publish the peer review history of their article (what does this mean?). If published, this will include your full peer review and any attached files.

Reviewer #1: **Yes: **Dr. Mahmoud Al-Masaeed

Reviewer #2: No

---

## [Editor Report · Acceptance letter]

8 Sep 2022

PONE-D-21-12441R1 

Associations of four indexes of social determinants of health and two community typologies with new onset type 2 diabetes across a diverse geography in Pennsylvania 

Dear Dr. Schwartz:

I'm pleased to inform you that your manuscript has been deemed suitable for publication in PLOS ONE. Congratulations! Your manuscript is now with our production department. 

Kind regards, 

on behalf of

Prof. Celine Rozenblat 

Academic Editor

PLOS ONE